# Special Attention to Physical Activity in Breast Cancer Patients during the First Wave of COVID-19 Pandemic in Italy: The DianaWeb Cohort

**DOI:** 10.3390/jpm11050381

**Published:** 2021-05-06

**Authors:** Valentina Natalucci, Milena Villarini, Rita Emili, Mattia Acito, Luciana Vallorani, Elena Barbieri, Anna Villarini

**Affiliations:** 1Department of Biomolecular Sciences, University of Urbino Carlo Bo, 61029 Urbino, Italy; valentina.natalucci@uniurb.it (V.N.); luciana.vallorani@uniurb.it (L.V.); 2Department of Pharmaceutical Sciences, University of Perugia, 06122 Perugia, Italy; mattia.acito@studenti.unipg.it; 3U.O.C. Oncologia Medica, ASUR Area Vasta 1, Ospedale Santa Maria della Misericordia di Urbino, 61029 Urbino, Italy; rita.emili@sanita.marche.it; 4Department of Research, Epidemiology Unit, Fondazione IRCCS Istituto Nazionale dei Tumori, 20133 Milano, Italy; anna.villarini@istitutotumori.mi.it

**Keywords:** breast cancer, COVID-19 pandemic, health management, physical activity, DianaWeb, epidemiology

## Abstract

Recent evidence highlights that physical activity (PA) is associated with decreased recurrence risk, improved survival and quality of life for breast cancer (BC) patients. Our study aimed to explore patterns of increased/decreased PA, and sedentary behaviors among BC women of the DianaWeb cohort during the first wave of COVID-19 pandemic, and examined the association with residential locations, work changes, different modality used to increase PA, and quality of life. The study analyzed the questionnaires completed by the 781 BC women (age 54.68 ± 8.75 years on both December 2019 and June 2020. Results showed a decrease of 22%, 57%, and 26% for walking activity, vigorous activity, and total PA, respectively. Sitting/lying time increased up to 54.2% of the subjects recruited. High quality of life was associated with lower odds of being sedentary (*p* = 0.003). Our findings suggest that innovative health management fostering compliance with current guidelines for PA and active behavior should be implemented, especially in unpredictable emergency conditions.

## 1. Introduction

Physical activity (PA) and exercise for breast cancer (BC) patients and survivors are emerging key elements in the oncological prevention spectrum. In this regard, exercise oncology (i.e., exercise medicine in the management of cancer) represents an important option for patients during rehabilitation, aftercare, and survival [1], with the aim of making the patient more active in everyday life. A growing body of literature shows the positive influence of PA and exercise on the reduction of recurrence and mortality [2,3]. Additionally, exercise can have a favorable impact on cancer- and treatment-related side effects (including fatigue, depression, and physical functioning) and quality of life (QoL) of cancer survivors. However, there are differences in outcomes depending on clinical setting of the BC patients and functional factors related to exercise, such as type, intensity, and activity level. Indeed, there is a positive correlation between high level of cardiorespiratory fitness and probability of survival [4], however, a high level of activity is not necessarily associated with the best QoL [5]. Ultimately, it is important to meet the stated American College of Sports Medicine (ACSM) recommendations on the basis of BC patient’s health status [6]. Indeed, current guidelines recommend people who have been treated for cancer to “avoid inactivity” and suggest that an effective exercise prescription includes moderate-intensity aerobic exercise at least three times per week. Moreover, the exercise program should add resistance-training activities, at least two times per week, using two sets of 8–15 repetitions at least 60% of one maximum repetition [6]. Unfortunately, population-based studies showed a general poor adherence to the PA guidelines in both the general population and cancer survivors, and data highlight that only 9–20% of the oncological patients meet both aerobic and resistance exercise guidelines, only 22–44% meet aerobic guidelines, and only 10–34% meet resistance guidelines [7].

The outbreak of the novel 2019 Coronavirus Disease (COVID-19) pandemic has represented a global public health emergency and routine cancer care, including health and supportive care interventions, was completely altered and movement behaviors have been impacted as well. Italy was the first European nation to be affected by COVID-19 which is, to date, a major global health issue. At the beginning of March 2020, the Italian Government adopted stringent containment measures on the entire national territory, which included lockdown and social distancing, to contain the spread of the virus SARS-CoV-2. The stringency of such measures has continuously varied [8], and also within the same country, according to the current diffusion of the disease and the burden on the healthcare system. In Italy, when the strictest measures have been adopted, the imperative was “stay at home”, to better control disease transmission, even at the cost of increasing risk factors for non-communicable diseases [9]. The policies and guidelines to implement physical distancing have significantly affected how people living with and beyond cancer spend their active time and receive cancer treatment. While the focus was mainly centered on cancer care and conventional standards in BC patients [10,11,12], little attention was paid to exercise oncology, although low levels of PA are recognized as an important risk factor.

The closure of common indoor and outdoor places to stay active, such as gyms, stadiums, pools, dance and fitness studios, physiotherapy centers, and parks and playgrounds, has undoubtedly had a negative impact on physiologic and psychosocial response of the general population [13], especially in people who have been diagnosed with BC and people who are at high risk for BC [14].

In this emergency context, it is possible that some BC women have altered their behaviors by facing additional barriers to PA, beyond those already documented [15]. Despite the challenges faced during this pandemic, we believe that it is important for BC women to continue to benefit from an active lifestyle in a safe environment.

The DianaWeb Project is a community-based participatory research that uses a specific interactive website which contributes to the growth of knowledge about lifestyles to be adopted by sharing recipes, movement strategies, and how to manage the change in daily practice involving Italian women with a BC diagnosis [16]. In this new scenario, understanding the barriers that may have influenced an active lifestyle could allow the development of further supportive strategies for oncological exercise.

In this study: (i) we described PA behavior of the DianaWeb cohort during the first wave of the COVID-19 pandemic, (ii) we made a comparison with data collected prior to lockdown, and (iii) we explored some factors that should be considered as moderators of PA, such as residential locations, living in an apartment building or in a dense living environment, BC clinical characteristics, or QoL, through private chat created for the study. Finally, we discussed the importance of identifying detrimental and positive lifestyle changes and the importance of developing possible interventions as an implementation of the DianaWeb platform for future PA coaching programs for women with BC.

## 2. Materials and Methods

DianaWeb protocol was previously detailed [17] and was approved by the ethics Committee of the Fondazione IRCCS Istituto Nazionale dei Tumori di Milano (Approval INT 24/16). Briefly, patients are recruited on a voluntary basis and, after having signed an informed consensus form, they are enrolled in the study. Once registered, all participants are requested to complete—twice a year—on-line questionnaires including: (a) the self-reported questionnaires on PA levels assessed by International Physical Activity Questionnaire Short Form (IPAQ-SF) [18,19], and (b) medical history. Participants also provided demographic information, anthropometric data (body weight, body height, and waist circumference), results of routine biochemical analysis, and clinical information (histology report and hospital discharge letters, and any other subsequent diagnosis). Volunteers can also make use of a private chat, supervised daily by researchers. Although dietary lifestyle habits are included in the DianaWeb platform, they are not a specific focus of this study.

### 2.1. Study Populations

Data was collected via an internet platform (http://www.dianaweb.org, accessed on July 2020). The DianaWeb is an open cohort established in September 2016. All Italian BC patients, whatever the disease stage at diagnosis, histological diagnosis, time elapsed since diagnosis, with or without metastasis, local recurrence or second cancers, and with in situ or invasive cancer, are eligible to join the cohort.

In particular, this study uses data collected on both December 2019 and June 2020. In December 2019, the DianaWeb cohort was composed of 1527 breast cancer women. Overall, we selected BC patients (*n* = 781) that completed the questionnaires in December 2019 and immediately after the first Italian lockdown (June 2020).

### 2.2. Questionnaires

The questionnaire areas are accessible only with patient ID and password. The questionnaires considered for this study provided the following data:(i)general information, such as sociodemographic characteristics (age, education level, marital status, region of residence, and residential density);(ii)anthropometric parameters (body weight, body height, and waist circumference);(iii)information about medical history (lymphedema arms, use of drugs, tumor metastasis, secondary tumor, etc.) and other health issues (from this section we collected information on SARS-CoV-2 positive swab);(iv)results of the last routine blood tests;(v)physical activity level, through the IPAQ-SF, whose reliability and validity are documented [18,19]: subjects reported the frequency (days/week) and duration (minutes/day) of different types of activity: vigorous (e.g., intense home or gardening activity, performing intense aerobic exercises, and using bike or treadmill); moderate (e.g., moderate home activity, work out in the garden, carrying light loads, and bicycling at a steady pace); and walking activities, as well as the average time spent sitting on a day; and(vi)lifestyle habits on QoL, through the question on one-dimension present in EORTC QLQ-C30 questionnaire [20]: global health-status/quality of life. The global health-status/quality of life scale has response options ranging from (1) “very poor” to (7) “excellent”.

In May 2020, participants freely provided information through the chat about: (a) different modality used to increase physical activity [technology-based interventions (e.g., apps, Facebook^®^, or Instagram); technology-based interventions with a personal trainer (e.g., video-conference, Skype, Zoom video communications including phone conversations, and FaceTime); non-technology interventions (autonomously, without technology support)]; (b) house dwelling floor space (e.g., <50 m^2^, 50–90 m^2^, and >90 m^2^) and private outdoor spaces (e.g., presence or absence of balcony and/or garden); (c) number of family members; and (d) working activity during quarantine.

### 2.3. Statistical Analysis

Frequency and percentage were provided for categorical data, whereas arithmetic means and standard deviation (SD) were provided for continuous variables. The patients were classified for residence as living in Northern, Central, or Southern Italy. Furthermore, by extending the analysis to residential density, the subjects were classified for living in cities, suburbs, or countryside cities. The education variable was dichotomized into high school or some college (≤13 years) and college graduates or higher (>13 years). The number of family members variable was trichotomized (1, 2, 3, or more members) as well as the dwelling floor space (<50 m^2^, 50–90 m^2^, and >90 m^2^).

PA levels were calculated from IPAQ-SF, converting questionnaire data in metabolic equivalent minutes per week (MET-min/week): each exercise intensity was associated with the metabolic equivalent of the task (MET): MET = 8 for vigorous, MET = 4 for moderate, MET = 3.3 for walking [21].

The BMI (kg/m^2^) was calculated using self-reported height and weight data. The degrees of obesity were established according to the World Health Organization’s (WHO) criteria: BMI: 18.5–24.9 kg/m^2^, normal weight; BMI: 25.0–29.9 kg/m^2^, overweight; BMI: 30.0–34.9 kg/m^2^, grade I obesity; BMI: 35.0–39.9 kg/m^2^, grade II obesity; and BMI ≥ 40.0 kg/m^2^, grade III obesity [22].

The χ^2^ test was used to compare qualitative data, whereas ANOVA was used to compare means of normally distributed quantitative data. In the case of statistically significant F-statistics, ANOVA was followed by a Dunnet post-hoc analysis. Pearson’s correlation coefficient was calculated to assess the strength and direction of the linear relationships between pairs of variables normally distributed. For non-ordinal variables, the Spearman correlation coefficient was calculated.

A linear multiple regression (LMR, block-wise) method was computed for PA levels (METs for moderate PA, vigorous PA, walking, and total PA) and sitting/lying time as dependent variables. Three blocks of variables were processed. Being the primary purpose of our LMR analysis was to explore the relationship between environment characteristics and PA, the first block consisted in area of residence, residential density, dwelling floor space, and private outdoor spaces. The second block contained socio-demographic variables (age, marital status, number of family members, level of education, and working activity). The third block was made up with health status variables (BMI, waist circumference, lymphedema, health perceptions, QoL, use of psychotropic drugs, and strategies to increase PA).

The independent variables that were relevant and significantly associated with PA from each block (*p* < 0.05) were included in the logistic regression analysis. Odds ratios values (OR = eβ), showing how the odds change with a one-unit increase in the independent variables, were also reported. All statistical analyses were carried out with SPSS software for Windows (version 20.0; SPSS Inc., Chicago, IL, USA) and *p*-values <0.05 were considered as statistically significant.

## 3. Results

### Sample Characteristics

Among the 1527 subjects enrolled in the DianaWeb cohort, 781 (51.5%) completed IPAQ-SF and EORTC-QLQ-C30 questionnaires on both December 2019 and June 2020.

Table 1 shows the main sociodemographic characteristics of women enrolled until December 2019 in the DianaWeb study, in particular, the whole cohort and subjects included (Group A) or not in the surveillance study (Group B).

Patients in the two sub-cohorts were similar for age (in both groups the enrolled women were in their 50s), marital status (most of the women were married), level of education, and Italian region of residence.

Table 2 presents the distribution of the study population also considering the presence of some barriers or facilitators for PA, as well as referring to the environment (population density, building design, and greenness), family (number of family members), working and clinical characteristics (lymphedema, and SARS-CoV-2 positive swab).

Almost half of the sample lived in cities with high population densities and in houses ≥ 90 m^2^ (55.1%) with one or more balconies (57.7%). Throughout the period covered by the study, only 9.2% of women worked outside of their homes; most of the women (45.1%) worked remotely.

From March until May 2020, 208 women (26.6%) of the DianaWeb surveillance were tested for SARS-CoV-2 infection and four of them resulted positive.

The anthropometric data after and before quarantine in subjects included in the surveillance study are presented in Table 3. Mean body weight, waist circumference (WC), and BMI were lower before than during quarantine. When individuals were categorized according to their WC (≤80 cm) or BMI (<18.5, 18.5–24.9 and ≥25.0), we did not observe any significant differences before and during quarantine.

In Table 4, results about QoL and health perception are reported. The analysis showed statistically significant differences between before and during quarantine for both parameters.

Stressful events may impact significantly on the initiation of psychotropic drug use. As Table 4 shows, the prevalence of psychotropic drugs use (such as anxiolytics, sedatives, and antidepressants) among participants was, during social isolation, about 16%.

In Table 5, the results about the PA section are reported. METs of walking, vigorous intensity, and total PA were significantly lower during quarantine, compared with before quarantine. The decreases during home confinement were about 22%, 57%, and 26%, respectively. Additionally, an increase was observed in sedentary behavior: daily sitting/lying time increased significantly from about 5 to 7 h/day, and during lockdown over 54% of women were high sitting (sitting more than 6 h/day).

The proportion of women who did vigorous PA or walking decreased significantly (Figure 1a). The proportion of women who were physically active (a combination of vigorous/moderate PA, and walking) decreased from 98.5% before quarantine to 93.7% during quarantine.

In addition, the IPAQ score expressed as MET-min/week was used as a general indicator of low active (MET < 600), moderate active (MET ≥ 600), and high active (MET ≥ 3000) people. We found an increase of low active women (<600 MET-min/week) from 9.3% before quarantine to 32.7% during quarantine, with a concurrent and significant reduction of high active women (≥3000 MET-min/week) from 24.7% to 11.4% (Figure 1b).

Participants most frequently indicated that they did PA without a gym instructor, and only 19.8% did PA with remote personal training (Figure 2).

The PA level and sitting/lying time, according to sociodemographic characteristics, barriers or facilitators to PA, self-reported PA strategies, anthropometric parameters, QoL, and health perception of the study population during quarantine are presented in Appendix A. A higher prevalence of physically active women was found among individuals which were 21–40 years old, separated or divorced, worked remotely, lived in Central Italy or in a large house with garden, did not use strategies to do PA, were underweight, did not suffer from lymphedema, and perceived their QoL and health as good.

In the multiple regression analysis (Table 6), block 1 showed that macroregion of residence and dwelling floor space were significant predictors of moderate PA. Based on our analysis, women living in Northern Italy or owning a house of 90 m^2^ or more resulted being facilitated in performing moderate PA. As reported in block 2, age and working activity were also significant predictors of moderate PA. METs from moderate PA were positively associated with age and with time spent at home (women who are retired or working at home increased their moderate PA). Age and number of family members had an inverse association with sitting or lying time. In block 3, BMI was negatively associated with walking, thus indicating that an increase in BMI may be associated with difficulty in walking. QoL had a negative association with sitting or lying time and a positive association with vigorous and total PA, showing that QoL is a key motivator of PA.

After identification of patterns involved in movement behavior changes during the first wave of the COVID-19 pandemic in Italy, we conducted logistic regression analysis (Table 7) including two built environment variables (microregion of residence and dwelling floor space), three socio-demographic variables (age, working activity, and number of family members), and one health status variable (QoL). Logistic regression models identified in QoL the independent variable that increased PA. The results indicated that women with higher values of QoL were more likely to increase vigorous PA (OR = 1.429; 95% CI 1.092–1.870), moderate PA (OR = 1.415; 95% CI 1.093–1.831), walking (OR = 1.432; 95% CI 1.211–1.693), and total PA (OR = 1.649; 95% CI 1.191–2.284). The logistic analysis showed that there were about 22% lower odds of sedentary (OR = 0.779; 95% CI 0.659–0.920; *p* = 0.003) for participants with high QoL, and 4% lower odds of sedentary (OR = 0.961; 95% CI 0.943–0.979; *p* = 0.001) for aged women.

## 4. Discussion

The DianaWeb study responds to the pressing request of patients diagnosed with BC to know the most advanced point of scientific research on the improvement of prognosis and to have a virtual space to meet, where to obtain evidence-based information about a healthy lifestyle [23]. The DianaWeb page can be effectively used to increase access to accurate information and to monitor participants’ lifestyles and health status over time in a very inexpensive way.

It has been observed that COVID-19 quarantine measures could have reduced PA and exercise in different subclasses of population [24,25,26], potentially causing various health side effects. Previous research has demonstrated that compared with individuals without a history of cancer, BC survivors are significantly more likely to develop unhealthy behaviors [27].

Our survey with 781 BC Italian women revealed that most of the participants reduced their PA level during the quarantine period, in which strict lockdown measures were adopted.

The results showed that MET-min/week of walking, vigorous intensity, and total PA were significantly lower during quarantine compared with before quarantine. In particular, the strongest differences were found in the percentage of high active women (from 24.7% before quarantine to 11.4% during quarantine) and sedentary women (from 9.3% before quarantine to 32.7% during quarantine). Our study also showed that during the pandemic, the daily sitting time significantly increased from about 5 to 7 h/day. Given that previous studies pointed out the detrimental effects of both sedentary behavior and PA on physical and psychological health [28], BC women of the whole lockdown sample were classified by time sitting/lying (h/day) (Table 5). During lockdown, more than 54% of the surveyed sample spent more than 6 h/day sitting. This phenomenon could be due to a radical change in everyday schedules and habits. However, to mitigate the deleterious effects of inactivity and social isolation, there are many creative ways to be physically active that do not require specialized technology and equipment. In this regard, our data showed that about 60% of the participants did PA autonomously, with non-technological interventions; about 40% had made use of technology-based interventions and only 20% had made use of technology-based interventions with a personal trainer. Although many suggestions and recommendations already exist [29] to PA practice, the COVID-19 pandemic highlighted the importance of understanding common barriers to PA practice and contrasting sedentary lifestyle, creating effective strategies in women with BC diagnosis. About that, this study showed that an emergency context influenced negatively the women’s PA behavior with an increase of 25% in inactive time and a decrease of 26% in active time, highlighting the importance of implementing cancer-management strategies.

Our survey agrees with other early reports on lifestyle habits during a pandemic and confirms that the quarantine restrictions were making people more sedentary than ever. In particular, an Italian study [30] highlighted that people who did not practice sports before the quarantine did not take advantage of this period as an opportunity to start and training frequency has increased only among those who already took part in sports. *Meyer et al.* [31] observed that in a sample of about 3000 American adults, people who were meeting exercise guidelines before the pandemic reported an average 32% reduction in PA level during the emergency and, interestingly, those who were sedentary before were inclined to keep their inactive condition [31].

The same behavior was observed in previously active BC survivors who reduced their PA, increased weight and sedentary behavior [32].

Intriguingly, the greatest prevalence of physically active women was among those aged 21 to 40, underweight, who did not suffer from lymphedema, and perceived their QoL and health as good. In multiple regression analyses, QoL was the only significant predictor for vigorous PA. Instead, macroregion of residence, dwelling floor space, age, and working activity were significant predictors of moderate PA. On the other side, age and number of family members had an inverse association with sitting or lying time. Notwithstanding, there were significant negative associations between sitting/lying time and age and QoL. We observed a slight increase in the use of psychotropic drugs (15.7% before vs. 16% during the quarantine) and, in accordance with our data, women with higher QoL values were more likely to increase total PA. These data support the positive association between exercise and improved physical/psychological health that has been well-established and demonstrated in people with cancer. In this regard, clinical evidence of exercise medicine efficacy in cancer management includes diminished symptomatology, enhanced functional capacity, and improved physical/psychological well-being, as well as a potential contribution to BC-specific mortality reduction, and possibly BC non-recurrence [33,34,35,36,37].

Efforts should be made to promote physical activity in BC patients. In this context, the implementation of the DianaWeb platform with specific coaching programs to overcome barriers, set realistic goals, and provide personalized advice adapted to BC patients can increase the proportion of women that meet the basic daily recommendation for the level of PA.

### Strengths and Limitations

The DianaWeb platform itself, centered on an interactive website (http://www.dianaweb.org, accessed on July 2020) designed to supervise the lifestyle habits and health status of BC patients and provide recommendations and suggestions for sustainable lifestyle changes, is considered an important strength. As a community-based participatory research, it is based on the collaborative involvement of all partners in all phases of the research, resulting in high compliance and an incisive knowledge dissemination process.

A limitation is the use of single items for unhealthy behaviors instead of more extensive measurement, e.g., devices to measure PA, which could have given a more precise estimate of the risk, as well as the self-reported questionnaire, which may lead to the actual misreporting of data.

One general limitation attributed to survey research is the oversimplification of social reality and the inconsistency of some collected data such as the percentage of COVID-19 infection within the DianaWeb Italian cohort.

## 5. Conclusions

This study showed that COVID-19 emergency increased the unhealthy behaviors in BC patients, indicative of a possible higher risk of worse prognosis. This observation was crucial to support our research group in improving the DianaWeb platform strategy.

In this context, the DianaWeb platform could help women with BC to maintain correct lifestyles based on continuous scientific information easily accessible through the internet, especially in those situations where it is harder to find and obtain conventional forms of professional communication. This tool might support clinical practice also through the development of smartphone apps that are more feasible and faster to use. Fitness applications for smartphones have enjoyed increasing popularity in recent years because of their ease of use. We intend to develop an app to track women’s dietary habits, how long they sleep, and how long they perform physical activity. This would be an intriguing way to collect data more objectively, in order to minimize memory bias related to self-compilation of questionnaires. Furthermore, in the future, tumor progression and/or survival data of the DianaWeb study participants will be traced to evaluate whether PA is able to reduce recurrence and mortality for BC. We have established a five-year follow-up to estimate the survival rate in the DianaWeb cohort and to compare it with BC survival rate in the Italian population.

## Figures and Tables

**Figure 1 jpm-11-00381-f001:**
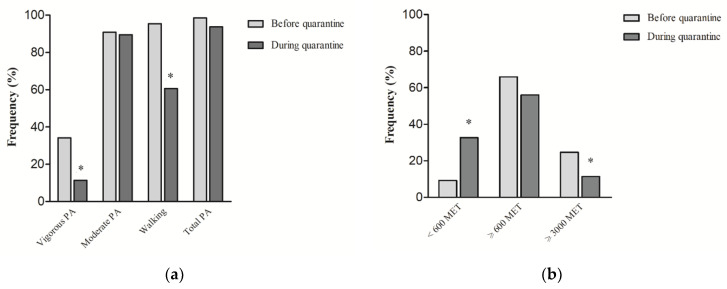
(**a**) Proportion of total and domain-specific physical activity of women before and during COVID-19 home confinement; (**b**) Proportion of low active (MET < 600), moderate active (MET ≥ 600), and high active (MET ≥ 3000) women before and during COVID-19 home confinement. * Before vs. during quarantine, χ2 test, *p* < 0.05; *Notes*: IPAQ score is expressed as MET-min/week; PA = Physical Activity.

**Figure 2 jpm-11-00381-f002:**
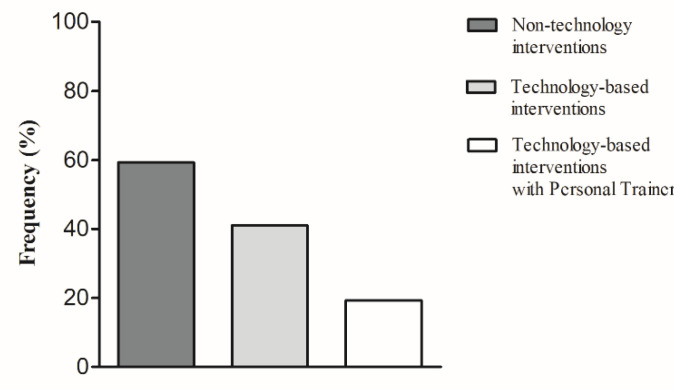
Proportion of self-reported physical activity modality of women before and during COVID-19 home confinement.

**Table 1 jpm-11-00381-t001:** Main sociodemographic characteristics of the DianaWeb cohort, and surveillance study participants.

Characteristic	Whole DianaWeb Cohort (*n* = 1527)	Group A (*n* = 781)	Group B (*n* = 746)	*p*
Age in years ^a^	54.14 (±8.80)	54.68 (± 8.75)	53.58 (±8.83)	0.014 ^d^
Young adults (aged 21–40) ^b^	85 (5.6)	37 (4.7)	48 (6.4)	0.170 ^e^
Adults (aged 41–60) ^b^	1.108 (72.6)	562 (72.0)	546 (73.2)	
Over 60 age ^b^	334 (21.9)	182 (23.3)	152 (20.4)	
Marital status ^b^				
Married	987 (64.6)	526 (67.3)	461 (61.8)	0.053 ^e^
Separated/divorced	177 (11.6)	92 (11.8)	85 (11.4)	
Widowed	44 (2.9)	21 (2.7)	23 (3.1)	
Never married	319 (20.9)	142 (18.2)	177 (23.7)	
Level of education ^b^				
High school or some college (≤13 years)	810 (53.0)	392 (50.2)	418 (56.0)	0.022 ^e^
College graduates or higher (>13 years)	717 (47.0)	389 (49.8)	328 (44.0)	
Region of residence ^b,c^				
Northern Italy	1033 (67.6)	576 (73.8)	457 (61.3)	0.000 ^e^
Central Italy	331 (21.7)	128 (16.4)	203 (27.2)	
Southern Italy	163 (10.7)	77 (9.9)	86 (11.5)	

^a^ Results expressed as the mean ± SD. ^b^ Results expressed as the number of subjects, percentage between brackets. ^c^ Northern Italy: Valle d’Aosta, Emilia-Romagna, Friuli-Venezia Giulia, Liguria, Lombardia, Piemonte, Trentino-Alto Adige, and Veneto. Central Italy: Lazio, Marche, Toscana, and Umbria. Southern Italy: Abruzzo, Puglia, Basilicata, Calabria, Campania, Molise, Sardegna, and Sicilia. ^d^ Group A vs. Group B, Student *t*-test. ^e^ Group A vs. Group B χ^2^ test.

**Table 2 jpm-11-00381-t002:** Facilitators or barriers to physical activity in subjects included in the surveillance study.

Facilitators or Barriers	Number of Subjects (%)
Residential density	
Cities	373 (57.7)
Suburbs	235 (30.1)
Countryside	173 (22.2)
House dwelling floor space	
<50 m^2^	45 (5.8)
50–90 m^2^	306 (39.2)
>90 m^2^	430 (55.1)
Private outdoor spaces	
None	48 (6.1)
Balcony	451 (57.7)
Garden	282 (36.1)
Number of family members	
1	224 (28.7)
2	248 (31.8)
3 or more	309 (39.6)
Working activity during quarantine	
Retired or laid off	215 (27.5)
Remote working	352 (45.1)
Normal working activity	72 (9.2)
Other	142 (18.2)
Lymphedema	
No	695 (89.0)
Yes	86 (11.0)
SARS-CoV-2 diagnostic test	
Positive	4 (0.5)
Negative	204 (26.1)
Not tested	573 (73.4)

**Table 3 jpm-11-00381-t003:** Anthropometric parameters in subjects included in the surveillance study.

	Before Quarantine	During Quarantine	*p*
Body Weight ^a^	61.46 ± 11.50	61.57 ± 11.03	0.525 ^c^
Waist circumference ^a^	80.52 ± 10.33	80.91 ± 11.03	0.101 ^c^
Normal ^b^	449 (57.5)	446 (57.1)	0.459 ^d^
Abdominal obesity ^b^	332 (42.5)	335 (42.9)	
Body mass index (BMI) ^a^	23.08 ± 4.00	23.13 ± 3.87	0.390 ^c^
Underweight ^b^	54 (6.9)	53 (6.8)	
Normal weight ^b^	542 (69.4)	528 (67.6)	0.678 ^d^
Overweight and obese ^b^	185 (23.7)	200 (5.6)	

^a^ Results expressed as the mean ± SD. ^b^ Results expressed as the number of subjects, percentage between brackets. ^c^ Before vs. during quarantine, student *t*-test. ^d^ Before vs. during quarantine, χ^2^ test.

**Table 4 jpm-11-00381-t004:** Quality of life and health perception in the studied population.

	Before Quarantine	During Quarantine	*p* ^b^
Quality of life ^a^			
Very poor	10 (1.3)	27 (3.5)	<0.001
Poor	44 (5.6)	146 (18.7)	
Neither poor nor good	238 (30.5)	306 (39.2)	
Good	421 (53.9)	275 (35.2)	
Very good	68 (8.7)	27 (3.5)	
Health perception ^a^			
Very poor	5 (0.6)	8 (1.0)	<0.001
Poor	37 (4.7)	113 (14.5)	
Neither poor nor good	253 (32.4)	273 (35.0)	
Good	423 (54.2)	341 (43.7)	
Very good	63 (8.1)	46 (5.9)	
Psychotropic drugs ^a^	123 (15.7)	128 (16.4)	0.391

^a^ Results expressed as the number of subjects, percentage between brackets. ^b^ Before vs. during quarantine, χ^2^ test.

**Table 5 jpm-11-00381-t005:** Level (MET-min/week assessed with IPAQ-SF score) and time sitting/lying (h/day) before and during nearly two months of quarantine.

	Before Quarantine	During Quarantine	Δ ^a^	*p*
Vigorous PA ^a^	361.95 ± 793.62	117.70 ± 468.78	−244.25 ± 685.82	<0.001 ^b^
Moderate PA ^a^	909.71 ± 902.68	888.53 ± 940.88	−21.18 ± 754.87	0.433 ^b^
Walking ^a^	941.22 ± 841.80	331.44 ± 590.33	−609.78 ± 801.77	<0.001 ^b^
Total PA ^a^	2212.87 ± 1696.11	1337.66 ± 1305.51	−875.20 ± 1361.51	<0.001 ^b^
Sitting time ≤ 6 h/day ^c^	480 (61.5)	358 (45.8)		<0.001 ^d^
Sitting time > 6 h/day ^c^	301 (38.5)	423 (54.2)		

^a^ Results expressed as the mean ± SD. ^b^ Before vs. during quarantine, student’s *t*-test. ^c^ Results expressed as the number of subjects, percentage between brackets. ^d^ Before vs. during quarantine, χ^2^ test. *Notes:* PA = Physical Activity.

**Table 6 jpm-11-00381-t006:** LMR between possible independent predictors and physical activity level (MET-min/week) or sitting/lying time (h/day) in the DianaWeb cohort during quarantine.

	Vigorous PA		Moderate PA		Walking		Total PA		Sitting/Lying	
	β	*p*	β	*p*	β	*p*	β	*p*	β	*p*
Block 1										
Region of residence	−0.032	0.380	0.076	0.032	0.031	0.392	0.057	0.107	0.004	0.905
Residential density	0.042	0.294	0.035	0.381	0.074	0.064	0.074	0.064	−0.043	0.280
Dwelling floor space	−0.011	0.780	0.092	0.015	−0.032	0.397	0.048	0.209	−0.060	0.115
Private outdoor spaces	−0.024	0.574	0.073	0.083	0.031	0.463	0.058	0.169	−0.068	0.105
Block 2										
Age	−0.074	0.070	0.097	0.017	0.058	0.155	0.070	0.090	−0.173	0.000
Marital status	−0.022	0.587	0.071	0.079	−0.008	0.837	0.039	0.334	0.038	0.349
Level of education	0.013	0.735	−0.031	0.407	0.073	0.052	0.015	0.681	0.045	0.221
Working activity	0.023	0.543	0.086	0.024	−0.024	0.523	0.059	0.122	−0.016	0.676
Family members	−0.011	0.788	0.072	0.082	0.008	0.846	0.052	0.218	−0.093	0.025
Block 3										
Body Mass Index	−0.046	0.402	0.039	0.484	−0.146	0.008	−0.055	0.322	0.067	0.227
Waist circumference	−0.024	0.666	0.010	0.860	0.074	0.174	0.032	0.561	−0.050	0.363
Lymphedema	0.012	0.731	−0.021	0.554	−0.030	0.401	−0.024	0.495	−0.037	0.302
Quality of life	0.098	0.034	0.070	0.135	0.217	0.000	0.184	0.000	−0.136	0.003
Health perception	−0.012	0.792	−0.017	0.715	−0.059	0.205	−0.044	0.352	0.028	0.559
Psychotropic drugs	−0.031	0.394	0.033	0.372	0.005	0.885	0.015	0.685	−0.011	0.772
Physical activity strategies	−0.003	0.931	−0.011	0.759	−0.025	0.486	−0.020	0.571	−0.004	0.912

**Table 7 jpm-11-00381-t007:** Logistic regression analysis between possible independent predictors and physical activity level (MET-min/week) or sitting/lying time (h/day) in the DianaWeb cohort during quarantine.

	B	*p*	OR	95% CI
Vigorous PA				
Region of residence	−0.176	0.335	0.839	0.587–1.199
Dwelling floor space	0.201	0.336	1.222	0.812–1.840
Age	−0.045	0.002	0.956	0.929–0.983
Working activity	−0.125	0.302	0.883	0.696–1.119
Family members	−0.040	0.796	0.961	0.710–1.301
Quality of life	0.357	0.009	1.429	1.092–1.870
Moderate PA				
Region of residence	0.033	0.861	1.033	0.714–1.495
Dwelling floor space	−0.010	0.962	0.990	0.655–1.496
Age	−0.006	0.674	0.994	0.965–1.024
Working activity	0.064	0.607	1.066	0.836–1.359
Family members	−0.138	0.392	0.871	0.636–1.194
Quality of life	0.347	0.008	1.415	1.093–1.831
Walking				
Region of residence	0.031	0.789	1.032	0.822–1.294
Dwelling floor space	−0.137	0.300	0.872	0.673–1.130
Age	0.017	0.081	1.017	0.998–1.036
Working activity	0.034	0.661	1.034	0.890–1.201
Family members	−0.032	0.747	0.968	0.796–1.178
Quality of life	0.359	0.000	1.432	1.211–1.693
Total PA				
Region of residence	0.017	0.945	1.017	0.637–1.624
Dwelling floor space	0.227	0.384	1.255	0.753–2.093
Age	−0.008	0.699	0.993	0.955–1.031
Working activity	−0.009	0.954	0.991	0.733–1.340
Family members	−0.255	0.217	0.775	0.517–1.162
Quality of life	0.500	0.003	1.649	1.191–2.284
Sitting/lying time				
Region of residence	−0.059	0.609	0.943	0.753–1.181
Dwelling floor space	−0.230	0.078	0.794	0.615–1.026
Age	−0.040	0.001	0.961	0.943–0.979
Working activity	0.029	0.704	1.029	0.887–1.193
Family members	−0.047	0.636	0.954	0.786–1.159
Quality of life	−0.250	0.003	0.779	0.659–0.920

*Notes:* PA = Physical Activity.

## Data Availability

The data presented in this study are available on request from the corresponding author.

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
