# Peer review of "Special Attention to Physical Activity in Breast Cancer Patients during the First Wave of COVID-19 Pandemic in Italy: The DianaWeb Cohort"

_jpm, 2021, doi:10.3390/jpm11050381_

Round 1

Reviewer 1 Report

This is an interesting study looking at the impact of PA changes in breast cancer patients in Italy. This is a well written paper with findings of clinical and public health significance. I do not have any comments.

Reviewer 2 Report

The manuscript does a good job of describing PA behavior of DianaWeb cohort during the first wave of COVID-19 pandemic and comparing with data collected prior to lockdown, and exploring factors that should be considered as moderators of PA. It also discusses the importance of identifying detrimental and positive lifestyle changes and the importance of developing possible interventions for future PA coaching programs for women with BC. This is a topic of great importance and the study design and analysis may benefit other research areas where similar analysis can be performed for other disease states.  Following are minor recommendations:

  • In the introduction, the statement, "A growing body of literature showed the positive influence of PA and exercise on the reduction of recurrence and mortality," includes no reference(s). Please cite a few references to support this statement.
  • Is there a way to track (in future) the progression and survival data on the participants that were part of this study? If this is possible, are the authors considering it as a possible future analysis. Since the ultimate role (and the positive impact) of PA is to reduce recurrence and mortality, it would be good to see what were the progression and survival rates in this cohort. If the authors plan (or not plan) such analysis in the future, please discuss this within this manuscript. 
